# Risk Factors for the Structural Progression of Myopic Glaucomatous Eyes with a History of Laser Refractive Surgery

**DOI:** 10.3390/jcm10112408

**Published:** 2021-05-29

**Authors:** Kwanghyun Lee, Bo Yi Kim, Gong Je Seong, Chan Yun Kim, Hyoung Won Bae, Sang Yeop Lee

**Affiliations:** 1Institute of Vision Research, Department of Ophthalmology, Yonsei University College of Medicine, Seoul 03722, Korea; ivynoel330@gmail.com (K.L.); BYKIMM@yuhs.ac (B.Y.K.); gjseong@yuhs.ac (G.J.S.); kcyeye@yuhs.ac (C.Y.K.); baekwon@yuhs.ac (H.W.B.); 2Department of Ophthalmology, National Health Insurance Service Ilsan Hospital, Goyang 10444, Korea; 3Department of Ophthalmology, Yongin Severance Hospital, Yonsei University College of Medicine, Yongin 16995, Korea

**Keywords:** glaucoma, myopia, laser refractive surgery, intraocular pressure, disc haemorrhage, optical coherence tomography, Goldmann applanation tonometry, LASIK, LASEK

## Abstract

As laser refractive surgeries (LRS) have been widely performed to correct myopia, ophthalmologists easily encounter patients with glaucoma who have a history of LRS. It is well known that intraocular pressure (IOP) in eyes with glaucoma is not accurate when measured using Goldmann applanation tonometry. However, risk factors for glaucoma progression, particularly those associated with measured IOP, have rarely been studied. We analysed data for 40 patients with a history of LRS and 50 age-matched patients without a history of LRS. Structural progression was defined as significant changes in thickness in the peripapillary retinal nerve fibre layer as identified using optical coherence tomography event-based guided progression analysis. Risk factors were determined via Cox regression analysis. Disc haemorrhage (DH) was associated with glaucoma progression in both the non-LRS group and LRS group (hazard ratio (HR): 4.650, *p* = 0.012 and HR: 8.666, *p* = 0.019, respectively). However, IOP fluctuation was associated with glaucoma progression only in the LRS group (HR: 1.452, *p* = 0.023). Our results show that DH was a significant sign of progression in myopic glaucoma eyes. When treating patients with myopia and glaucoma, IOP fluctuation should be monitored more carefully, even if IOP seems to be well controlled.

## 1. Introduction

Laser refractive surgeries (LRSs), such as photorefractive keratectomy, laser in situ keratomileusis (LASIK), and laser in situ epithelial keratomileusis (LASEK), are widely performed to correct myopia. Since myopia is a well-known risk factor for glaucoma [1,2], ophthalmologists commonly encounter patients with glaucoma who have a history of LRS or are planning to undergo LRS. Patients with glaucoma often have some concerns regarding the effects of LRS on the development or progression of glaucoma.

Several studies have investigated whether LRS exacerbates glaucomatous damage or aggravates glaucoma progression. Acute increases in intraocular pressure (IOP) can damage the optic nerve during surgical procedures [3], and studies have also reported that topical steroid use can cause postoperative increases in IOP [4,5]. Another concern is the reliability of IOP measurements after LRS. Since IOP is the most important risk factor for glaucoma progression [6,7], accurate IOP measurement is critical for glaucoma management. However, LRS changes the central corneal thickness (CCT) and corneal curvature, which consequently affect IOP measured via Goldmann applanation tonometry (GAT), [8,9] resulting in erroneous IOP measurements. Because of these inaccurate IOP measurements, glaucoma progression may be more common in eyes with a history of LRS than in those without a history of LRS, and IOP management in glaucomatous eyes with a history of LRS may require further investigation.

Despite these concerns, few studies have investigated the effects of IOP on glaucoma progression in eyes with a history of LRS. Although one previous study reported risk factors for glaucoma progression in eyes that had undergone LRS [10], the effects of IOP measured during the follow-up period were not investigated. Therefore, in this study, we aimed to investigate the effects of clinical variables, including IOP measurements, on structural progression in glaucomatous eyes with a history of LRS and compare them to age-matched glaucomatous eyes without a history of LRS. We used optical coherence tomography (OCT) to investigate structural progression, as it is considered more sensitive to the detection of progression in the early stages of the disease that may be not detected via visual field (VF) tests [11,12,13]. 

## 2. Materials and Methods

This retrospective cohort study was approved by the Institutional Review Board of Yongin Severance Hospital (9-2020-0105) and adhered to the tenets of the Declaration of Helsinki. We retrospectively reviewed the medical records of all patients who visited the glaucoma clinic at Severance Hospital from February 2011 to January 2020. The requirement for informed consent was waived because of the retrospective study design, and all clinical data were anonymised.

Inclusion criteria were as follows: an open-angle on gonioscopy, best-corrected visual acuity >20/30, no medical history of systemic disease, and no history of anti-glaucoma medication use before the initial clinic visit. At least eight IOP measurements using GAT at regular clinic visits were required for inclusion. Patients with glaucoma who underwent ophthalmic surgeries other than LRS (e.g., cataract surgery, vitrectomy, or glaucoma surgery) were excluded. Patients who discontinued using anti-glaucoma medication during the follow-up period(s) and those who used steroid or non-steroidal anti-inflammatory eye drops were also excluded. To minimise the floor effect of OCT measurements [14], we excluded eyes with a mean deviation (MD) of <−20 dB or peripapillary retinal nerve fibre layer (RNFL) thickness of <60 μm. If both eyes were eligible for inclusion, one eye was randomly selected.

All participants underwent complete ophthalmic examinations: measurements of best-corrected visual acuity, IOP using GAT, CCT measurements using an ultrasonic pachymeter (DGH-1000; DGH Technology, Inc., Frazer, PA, USA), slit-lamp biomicroscopy, dilated fundus examination, colour disc photography (Carl Zeiss Meditec, Jena, Germany), spectral-domain OCT (Cirrus HD-OCT, software v11.0; Carl Zeiss Meditec), axial length measurements (IOL Master; Carl Zeiss Meditec), and VF tests (Humphrey Field Analyzer II; Carl Zeiss Meditec).

The diagnosis of glaucoma was based on the characteristics of glaucomatous optic disc change and RNFL defect with VF defect. VF defects had to satisfy at least two of the Anderson criteria [15]. Two glaucoma specialists (S.Y.L. and K.L.) reviewed the medical records of patients with glaucoma. In case of disagreements between these specialists, a third glaucoma specialist (C.Y.K.) was consulted to verify the diagnosis.

IOP was measured at each follow-up visit in accordance with the standard procedure of our institution [16]. IOP measurements during follow-up periods were averaged (mean IOP), and the maximum IOP value (peak IOP) was extracted. To manage IOP fluctuations, the standard deviation of all IOP values measured during the follow-up period was calculated (IOP fluctuation). Adjustments of IOP were made using the following linear formula: Corrected IOP= Measured IOP − (CCT − 545)/2.5 mm Hg [17].

The LRS group included individuals who underwent LRS surgery, including LASIK and LASEK, at least 3 years before the initial visit to the clinic [16]. Participants with myopia (refractive error <−0.5 D or axial length ≥24.0 mm) without a history of LRS and younger than 60 years were categorised into the non-LRS group [16]. All patients with LRS participated in the study when myopia was fully corrected by LRS, except those who experienced an IOP elevation after LRS. Because our preliminary data showed that the LRS group was younger than the non-LRS group, we only included individuals younger than 60 years to minimise the effect of age on structural progression. 

### 2.1. Optical Coherence Tomography and Guided Progression Analysis

OCT images of the peripapillary RNFL were obtained with an optic disc cube scan using Cirrus HD-OCT. The optic disc cube scan produced an RNFL thickness map of 6 × 6 mm (200 × 200 pixels) in the area centred on the optic nerve head. Peripapillary RNFL thickness was measured circularly with a diameter of 3.46 mm. At least five reliable OCT scans from separate visits were required for inclusion in the study. All OCT scans had a signal strength of ≥6. Scans with motion artefacts, poor centration, or missing data were excluded. All scans were reviewed by glaucoma specialists (S.Y.L. and K.L.). If peripapillary RNFL was measured inaccurately due to myopic disc morphology (optic disc torsion or tilt), those scans were excluded from the analysis.

We evaluated the structural progression of the peripapillary RNFL using an event-based algorithm provided by guided progression analysis. The guided progression analysis algorithm compared the changes in peripapillary RNFL thickness at individual superpixels (1 superpixel = 4 × 4 pixels) between two baseline thickness maps and the follow-up thickness map. For the change to have been considered significant, a change of at least 20 adjacent superpixels must have been detected in the RNFL thickness maps. If a follow-up OCT examination demonstrated a statistically significant difference in thickness that exceeded the baseline test–retest variability, the superpixel was labelled yellow to indicate a possible loss. If confirmed on a second follow-up OCT examination, it was labelled red to indicate a probable loss. Structural progression was defined as the detection of “likely loss” in the event analysis, and this change was observed at the most recent follow-up visit.

### 2.2. Statistical Analyses

Statistical analyses were performed using R software v3.6.3 (R Foundation for Statistical Computing, Vienna, Austria). Baseline clinical variables are presented as the mean ± standard deviation. We used *t*-tests to compare variables between the non-LRS and LRS groups and between stable eyes and eyes with structural progression in each group. The chi-square test was used to compare categorical data. Hazard ratios (HRs) of potential risk factors for structural progression were investigated using Cox regression analysis. Variables with a *p*-value < 0.05 in the univariate analysis were considered significant and were included in a multivariate Cox regression analysis. All tests reported *p*-values as bilateral; those less than 0.05 were considered statistically significant.

## 3. Results

A total of 90 glaucomatous eyes of 90 individuals were included in our study. Among them, 40 eyes had a history of LRS, and 50 age-matched eyes had no history of LRS. All patients were treated with an anti-glaucoma medication. The mean age of the study population was 42.5 ± 10.7 years. The mean baseline peripapillary RNFL thickness was 79.0 ± 10.7 µm, and the average of VF MD was −4.8 ± 4.5 dB. A summary of the other variables is presented in Table 1. 

The occurrence of structural progression was not different between the non-LRS and LRS groups (Table 1). The baseline peripapillary RNFL thickness, VF MD, sex, and history of hypertension and diabetes did not differ between the two groups (Table 1). However, baseline IOP, mean IOP, disc haemorrhage (DH), and CCT were significantly different between the two groups, although IOP fluctuation and peak IOP were not (Table 1). 

In the non-LRS group, patients with glaucoma progression were younger and had better MD than those without glaucoma progression. In the LRS group, patients with glaucoma progression had a thicker baseline peripapillary RNFL and a higher mean IOP and peak IOP than those without glaucoma progression (Table 2).

Cox regression results are shown in Table 3 and Table 4. In the non-LRS group, young age, history of DH, and better baseline MD were associated with glaucoma progression. Multivariate Cox regression analysis revealed that age and history of DH were related to structural progression (Table 3). In the LRS group, a history of DH and a larger IOP fluctuation were associated with structural progression in the univariate and multivariate Cox analyses (Table 4).

## 4. Discussion

In this study, we investigated the risk factors for structural progression in glaucomatous eyes with and without a history of LRS. Although the occurrence of progression did not significantly differ between the non-LRS and LRS groups, risk factors differed between the two groups. According to the results of the multivariate Cox regression analysis, DH was related to structural progression, regardless of a history of LRS. Long-term IOP fluctuation was associated with structural progression only in the LRS group, whereas age was associated with structural progression only in the non-LRS group. 

Cox regression analysis revealed that IOP fluctuation was associated with structural progression in the LRS group, whereas mean IOP and peak IOP were not. The relationship between long-term IOP fluctuation and glaucoma progression has been controversial [18,19,20,21]. Our results are consistent with those of the Advanced Glaucoma Intervention Study (AGIS), which reported that long-term IOP fluctuation is the most critical risk factor for VF progression in glaucoma [20]. A large retrospective study of patients treated for primary open-angle glaucoma or primary angle-closure glaucoma also reported that IOP fluctuation is significantly associated with disease progression [19]. However, the Early Manifest Glaucoma Trial (EMGT) reported that mean IOP is related to disease progression, while IOP fluctuation is not [18]. These conflicting results may be due to the differences in the mean IOP of the study population. The participants of the EMGT, including untreated patients with early glaucoma, had a mean IOP of 20.7 ± 4.1 mmHg, which was greater than that in our study (13.4 ± 2.4 mmHg) and the AGIS. These results imply that a threshold mean IOP is required to induce glaucoma progression. The mean IOP of our study participants was controlled to be less than 15 mmHg; thus, the effect of the mean IOP on glaucoma progression may be insignificant. Our results do not suggest that the mean IOP is not related to structural progression but that when the patient’s mean IOP is well controlled, structural progression could occur due to IOP fluctuations. 

Interestingly, IOP fluctuation was a risk factor for structural progression only in the LRS group, even though it did not significantly differ between the two groups. While the relationship between long-term IOP fluctuation and glaucoma progression remains controversial [18,19,20,21,22], a recent study reported that long-term IOP fluctuation could be a significant risk factor for structural progression in medically treated glaucoma eyes with a mean IOP of ≤15 mmHg [14]. Since the mean IOP of our participants was 13.3 mmHg, IOP fluctuation could show the association with glaucoma progression. Our result implies that IOP fluctuation, as with IOP measurements, may be underestimated in the LRS group. We previously reported on the difference between IOP measured via GAT and IOP measured via dynamic contour tonometry (DCT) in LRS eyes. In that study, the fluctuation of IOP measured via DCT was 1.93 mmHg, which was higher than that measured via GAT (1.77 mmHg). Considering that DCT measurements are known to better reflect the actual IOP [23], the actual IOP fluctuation in the LRS group is assumed to be larger than that calculated from GAT measurements. Moreover, our results suggest that IOP fluctuation could be closely related to a structural progression in myopic glaucomatous eyes. A recent study has reported that long-term IOP fluctuation is more significantly related to VF progression in myopic normal-tension glaucoma (NTG) eyes than in non-myopic NTG eyes [24]. Myopic eyes are known to have a thinner lamina cribrosa than non-myopic eyes [25]; thus, they may be more vulnerable to the same degree of IOP fluctuation than non-myopic eyes. In addition, recent studies reported that 24-h IOP fluctuation was associated with VF progression and axial length. Therefore, further studies that include 24 h IOP profile analysis are recommended. 

DH was reaffirmed as a well-known risk factor for NTG progression [26] in this study. We included patients with glaucoma, regardless of baseline IOP. The mean baseline IOP of our participants was 16.0 ± 3.3 mmHg, suggesting that most participants were patients with NTG. DH has been considered to indicate progressive structural damage in glaucomatous eyes. That is, optic disc changes, new RNFL defects, and enlargement of pre-existing RNFL defects are more likely to occur after DH [26,27,28]. These results are consistent with our finding that DH is a risk factor for structural progression. Although the finding that DH is closely related to structural progression has been reported consistently in numerous studies, the reported range of HR values is large. In our study, the HRs of DH for structural progression were 4.650 in the non-LRS group and 8.666 in the LRS group, similar to those in a previous study that reported HRs of 15.533 in median NTG eyes and 7.596 in conventional NTG eyes [2]. In contrast, Kim et al. [29] reported that the HR of DH was 1.718 when analysing 127 patients with pre-perimetric open-angle glaucoma. These different results may be attributable to the different characteristics of the participants. Lee et al. [2] reported that DH is a risk factor in conventional NTG eyes but not in conventional high-tension glaucoma (HTG) eyes. When the study participants were divided according to the median IOP, the HR of DH in median NTG eyes (median IOP ≤15 mmHg) was 13.926, which was higher than the HR (7.572) in median HTG eyes (median IOP >15 mmHg). Overall, these results indicate that the relationship between DH and glaucoma progression may be stronger in NTG eyes. 

One of the unexpected findings of our study was the relationship between younger age and glaucoma progression found in the non-LRS group. According to previous studies, older age increases the risk of glaucoma progression [6,30]. These contradictory results may be attributable to treatment selection bias. Clinicians who know that glaucoma can progress faster in older patients may attempt to control patients’ IOP more strictly. In our study, the mean IOP in the non-LRS group tended to decrease with age (data now shown). In other words, the mean IOP of younger patients in the non-LRS group was relatively high, which may have enabled the detection of glaucoma progression in these young patients.

CCT was not associated with structural progression in our study, but previous studies have reported an association between CCT and glaucoma progression [31,32]. The results of the Ocular Hypertension Treatment Study (OHTS) demonstrated that CCT is an important risk factor for glaucoma progression in patients with ocular hypertension [32], but the EMGT study failed to find a significant association between CCT and glaucoma progression [33]. These results may be explained by differences in baseline IOP between studies. Based on the OHTS results, the effect of CCT on glaucoma progression was stronger in eyes with IOP above 25.75 mmHg, suggesting that the effect of CCT on glaucoma progression is stronger in those with higher IOP. In our study, the mean IOP of the participants (13.5 ± 2.3 mmHg) was similar to that in the EMGT study and lower than that in the OHTS; thus, a significant association between CCT and glaucoma progression may not have been observed. Several recent studies have reported corneal biomechanical variables besides CCT (e.g., low corneal hysteresis) that are associated with glaucoma progression [34,35]; therefore, further study regarding this relationship is necessary. 

This study has several limitations. First, this study included a small number of patients. To ensure sufficient statistical power, a previous study was referenced [14], and the statistical power was confirmed through a post hoc analysis. The required sample size was at least 26 when calculated using mean IOP and structural progression in the LRS group. However, some variables may need to be analysed with a larger sample size to have enough statistical power to show a significant effect, especially in the non-LRS group. Second, we analysed structural progression only, without considering functional progression. Because OCT is not sensitive for detecting progression in patients with advanced glaucoma, eyes with advanced glaucoma were excluded from the analyses. Therefore, the risk factors identified in this study may not apply to advanced glaucoma. Additionally, only peripapillary RNFL was analysed to detect structural progression in this study. One recent study reported that monitoring of macular ganglion cell-inner plexiform layer measurements could be effective for predicting glaucoma progression in high myopia [36]. Therefore, we believe that further studies that include macular ganglion cell-inner plexiform layer analysis are needed. Third, because the baseline IOP of the study participants was in the NTG range, the results of our study may not be applicable to patients with open-angle glaucoma whose baseline IOP is not within the normal range. Our finding that IOP fluctuation, not mean or peak IOP, was significantly related to structural progression may be limited to patients with NTG. Therefore, further studies including patients with open-angle glaucoma with a higher baseline IOP are necessary. Forth, due to the retrospective nature of this study, IOP was measured by different examiners. The accuracy of IOP measurements may be affected by the inter-examiner differences. Despite this limitation, the IOP of the participants in this study were measured using a consistent and standardised method, so inter-examiner differences for IOP measurements were considered controlled. 

## 5. Conclusions

Risk factors for structural progression of open-angle glaucoma differed between the LRS and non-LRS groups. IOP fluctuation and DH were significant risk factors in glaucomatous eyes with a history of LRS. These results suggest that when managing glaucoma patients with a history of LRS, IOP fluctuation should be minimised, and more aggressive treatment should be considered if DH is found.

## Figures and Tables

**Table 1 jcm-10-02408-t001:** Demographic characteristics.

	Total	Non-LRS	LRS	*p*-Value
	(*n* = 90)	(*n* = 50)	(*n* = 40)	
Age (years)	42.5 ± 10.7	42.7 ± 12.0	42.2 ± 8.9	0.825
Sex (male/female)	44 (48.9%)/46 (51.1%)	25 (50.0%)/25 (50.0%)	19 (47.5%)/21 (52.5%)	0.981
Hypertension (Y/N)	11 (12.2%)/79 (87.8%)	4 (8.0%)/46 (92.0%)	7 (17.5%)/33 (82.5%)	0.297
Diabetes (Y/N)	4 (4.4%)/86 (95.6%)	2 (4.0%)/48 (96.0%)	2 (5.0%)/38 (95.0%)	0.999
Disc haemorrhage (Y/N)	18 (18.6%)/79 (81.4%)	13 (26.0%)/37 (74.0%)	3 (7.5%)/37 (92.5%)	**0.045**
Baseline RNFL thickness (µm)	79.0 ± 10.7	79.3 ± 10.2	78.6 ± 11.4	0.768
MD (decibel)	−4.8 ± 4.5	−4.3 ± 4.4	−5.5 ± 4.5	0.191
Axial length (mm)	26.4 ± 1.1	26.0 ± 1.1	26.6 ± 1.1	0.051
Central corneal thickness (µm)	507.2 ± 50.2	539.4 ± 27.7	470.9 ± 44.9	**<0.001**
Baseline IOP (mmHg)	16.0 ± 3.3	16.6 ± 3.1	14.7 ± 3.5	**0.020**
Mean IOP (mmHg)	13.3 ± 2.5	14.3 ± 2.1	12.0 ± 2.4	**<0.001**
IOP fluctuation (mmHg)	1.9 ± 1.0	1.7 ± 0.6	2.1 ± 1.4	0.076
Peak IOP (mmHg)	16.8 ± 4.0	17.2 ± 2.8	16.2 ± 5.2	0.279
Adjusted baseline IOP (mmHg)	20.2 ± 3.4	21.0 ± 3.1	18.9 ± 3.5	**0.014**
Adjusted mean IOP (mmHg)	17.4 ± 2.6	18.5 ± 2.2	16.2 ± 2.4	**<0.001**
Adjusted Peak IOP (mmHg)	21.0 ± 4.1	21.4 ± 2.9	20.4 ± 5.2	0.267
Structural progression (stable/progressed)	54 (60.0%)/36 (40.0%)	29 (58.0%)/21 (42.0%)	25 (62.5%)/15 (37.0%)	0.829
Follow-up period (months)	60.3 ± 18.2	58.4 ± 18.0	62.6 ± 18.6	0.287

Parameters are presented as the mean ± standard deviation or *n* (%). LRS: laser refractive surgery; Y: yes; N: no; RNFL: retinal nerve fibre layer; MD: mean deviation; IOP: intraocular pressure. *p*-values were calculated using *t*-tests (comparison between the non-LRS and LRS groups). Indicated in bold type, *p* < 0.05 indicates statistical significance.

**Table 2 jcm-10-02408-t002:** Comparison of variables between stable eyes and eyes with structural progression in the non-LRS and LRS groups.

	Non-LRS			LRS		
Structural Progression	Stable	Progressed	*p*-Value	Stable	Progressed	*p*-Value
	(*n* = 29)	(*n* = 21)		(*n* = 25)	(*n* = 15)	
Age (years)	47.3 ± 11.6	36.3 ± 9.6	**0.001**	42.8 ± 8.1	41.1 ± 10.3	0.548
Sex (male/female)	16 (55.2%)/13 (44.8%)	9 (42.9%)/12 (57.1%)	0.567	13 (52.0%)/12 (48.0%)	6 (40.0%)/9 (60.0%)	0.683
Hypertension (Y/N)	2 (6.9%)/27 (93.1%)	2 (9.5%)/19 (90.5%)	0.999	4 (16.0%)/21 (84.0%)	3 (20.0%)/12 (80.0%)	0.999
Diabetes (Y/N)	1 (3.4%)/28 (96.6%)	1 (4.8%)/20 (95.2%)	0.999	2 (8.0%)/23 (92.0%)	0 (0.0%)/15 (100.0%)	0.708
Disc haemorrhage (Y/N)	6 (20.7%)/23 (79.3%)	7 (33.3%)/14 (66.7%)	0.497	1 (4.0%)/24 (96.0%)	2 (13.3%)/13 (86.7%)	0.642
Baseline RNFL thickness (µm)	77.3 ± 10.6	82.1 ± 9.1	0.099	74.6 ± 9.8	85.3 ± 11.1	**0.003**
MD (dB)	−5.9 ± 5.1	−2.1 ± 1.6	**0.001**	−6.4 ± 4.5	−4.0 ± 4.2	0.107
Axial length (mm)	25.8 ± 0.7	26.3 ± 1.3	0.255	26.7 ± 1.2	26.5 ± 1.1	0.746
Central corneal thickness (µm)	542.3 ± 26.6	535.1 ± 29.5	0.397	469.6 ± 46.7	473.3 ± 43.4	0.804
Baseline IOP (mmHg)	16.5 ± 3.4	16.8 ± 2.7	0.746	14.7 ± 3.6	14.8 ± 3.7	0.956
Mean IOP (mmHg)	14.1 ± 2.0	14.6 ± 2.3	0.400	11.2 ± 2.0	13.5 ± 2.5	**0.014**
IOP fluctuation (mmHg)	1.6 ± 0.4	1.8 ± 0.7	0.211	1.7 ± 0.9	2.7 ± 1.8	0.069
Peak IOP (mmHg)	16.9 ± 2.2	17.6 ± 3.4	0.404	14.8 ± 4.5	18.6 ± 5.6	**0.023**
Adjusted baseline IOP (mmHg)	20.8 ± 3.4	21.3 ± 2.7	0.604	18.9 ± 3.6	18.9 ± 3.7	0.956
Adjusted mean IOP (mmHg)	18.3 ± 2.0	18.7 ± 2.4	0.537	15.4 ± 2.0	17.7 ± 2.5	**0.002**
Adjusted Peak IOP (mmHg)	21.2 ± 2.2	21.9 ± 3.6	0.477	19.0 ± 4.5	22.8 ± 5.6	**0.023**

The parameters are presented as the mean ± standard deviation or *n* (%). LRS: laser refractive surgery; Y: yes; N: no; RNFL: retinal nerve fibre layer; MD: mean deviation; IOP: intraocular pressure. *p*-values were calculated using *t*-tests (comparison between stable eyes and progressed eyes in each group). Indicated in bold type, *p* < 0.05 indicates statistical significance.

**Table 3 jcm-10-02408-t003:** Cox regression for the non-LRS group.

	Univariate		Multivariate	
	HR	*p*-Value	HR	*p*-Value
Age (years)	0.935 (0.892–0.979)	0.004	0.921 (0.868–0.977)	0.007
Sex (male/female)	1.369 (0.573–3.269)	0.479		
Hypertension (Y/N)	0.904 (0.208–3.929)	0.893		
Diabetes (Y/N)	0.766 (0.099–5.902)	0.798		
Disc haemorrhage (Y/N)	2.664 (1.053–6.742)	0.039	4.650 (1.402–15.416)	0.012
Baseline RNFL thickness (µm)	1.024 (0.989–1.061)	0.181		
MD (decibel)	1.248 (1.049–1.486)	0.013	1.097 (0.929–1.295)	0.277
Axial length (mm)	0.901 (0.526–1.546)	0.706		
Central corneal thickness (µm)	0.998 (0.980–1.017)	0.846		
Baseline IOP (mmHg)	0.934 (0.808–1.079)	0.353		
Mean IOP (mmHg)	0.983 (0.795–1.216)	0.875		
IOP fluctuation (mmHg)	0.842 (0.391–1.811)	0.660		
Peak IOP (mmHg)	0.950 (0.811–1.111)	0.520		
Adjusted baseline IOP (mmHg)	0.961 (0.823–1.121)	0.611		
Adjusted mean IOP (mmHg)	1.021 (0.827–1.262)	0.884		
Adjusted Peak IOP (mmHg)	0.983 (0.845–1.145)	0.829		

LRS: laser refractive surgery; HR: hazard ratio; Y: yes; N: no; RNFL: retinal nerve fibre layer; MD: mean deviation; IOP: intraocular pressure. Indicated in bold type, *p* < 0.05 indicates statistical significance.

**Table 4 jcm-10-02408-t004:** Cox regression for the LRS group.

	Univariate		Multivariate	
	HR	*p*-Value	HR	*p*-Value
Age (years)	1.001 (0.942–1.065)	0.965		
Sex (male/female)	1.006 (0.319–3.173)	0.992		
Hypertension (Y/N)	1.339 (0.361–4.963)	0.662		
Diabetes (Y/N)	NA (0.000–Infinite)	0.999		
Disc haemorrhage (Y/N)	6.144 (1.123–33.611)	**0.036**	8.666 (1.431–52.480)	**0.019**
Baseline RNFL thickness (µm)	1.037 (0.994–1.082)	0.090		
MD (decibel)	1.023 (0.902–1.160)	0.723		
Axial length (mm)	0.846 (0.480–1.490)	0.562		
Central corneal thickness (µm)	1.000 (0.988–1.011)	0.938		
Baseline IOP (mmHg)	1.041 (0.870–1.244)	0.662		
Mean IOP (mmHg)	1.173 (0.961–1.432)	0.117		
IOP fluctuation (mmHg)	1.357 (1.001–1.839)	**0.050**	1.452 (1.052–2.004)	**0.023**
Peak IOP (mmHg)	1.061 (0.969–1.162)	0.201		
Adjusted baseline IOP (mmHg)	1.041 (0.870–1.244)	0.662		
Adjusted mean IOP (mmHg)	1.173 (0.961–1.432)	0.117		
Adjusted Peak IOP (mmHg)	1.061 (0.969–1.162)	0.201		

LRS: laser refractive surgery, HR: hazard ratio; Y: yes; N: no; RNFL: retinal nerve fibre layer; MD: mean deviation; NA: not available; IOP: intraocular pressure. Indicated in bold type, *p* < 0.05 indicates statistical significance.

## Data Availability

The data that support the findings of this study are available from the corresponding author (S.Y.L.) upon reasonable request.

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
