# Peer review of "Risk Factors for the Structural Progression of Myopic Glaucomatous Eyes with a History of Laser Refractive Surgery"

_jcm, 2021, doi:10.3390/jcm10112408_

Round 1

Reviewer 1 Report

The authors analyzed data for 40 patients with a history of LRS and 50 age-matched patients without a history of LRS and they found that DH was a significant sign of progression in myopic glaucoma eyes. When treating patients with myopia and glaucoma, IOP fluctuation should be monitored more carefully, even if IOP seems to be well controlled.

  1. The manuscript is well written, and the topic is interesting to ophthalmologist scientific community, some point must be addressed.
  2. Purpose or objective at the end of the introduction is missing
  3. We found that at the end of the intro the authors stated some summary results and kind of a conclusion, remove that, and changed by a purpose.
  4. Inclusion and exclusion criteria must be stated prior to previous examination
  5. Provide on discussion some clinical practice advises to improve this technology in dairy ophthalmology routine.
  6. Please clarify the conclusion to improve comprehension. Resume into 3 or 4 bullets sentence and provided shortened sentences.

Author Response

The authors analyzed data for 40 patients with a history of LRS and 50 age-matched patients without a history of LRS and they found that DH was a significant sign of progression in myopic glaucoma eyes. When treating patients with myopia and glaucoma, IOP fluctuation should be monitored more carefully, even if IOP seems to be well controlled.

The manuscript is well written, and the topic is interesting to ophthalmologist scientific community, some point must be addressed.

-> We appreciate your positive feedback and hope that the revised manuscript is now suitable for publication.

1. Purpose or objective at the end of the introduction is missing

2. We found that at the end of the intro the authors stated some summary results and kind of a conclusion, remove that, and changed by a purpose.

-> We edited for the objective of this study to be located at the end of the introduction.

-> Lines 50-56. Despite these concerns, few studies have investigated the effects of IOP on glaucoma progression in eyes with a history of LRS. Although one previous study reported risk factors for glaucoma progression in eyes that had undergone LRS [10], the effects of IOP measured during the follow-up period were not investigated. Therefore, in this study, we aimed to investigate the effects of clinical variables, including IOP measurements, on structural progression in glaucomatous eyes with a history of LRS and compare them to age-matched glaucomatous eyes without a history of LRS.

3. Inclusion and exclusion criteria must be stated prior to previous examination

-> We moved the paragraph about inclusion and exclusion criteria as recommended.

-> Lines 68-85. Inclusion criteria were as follows: an open-angle on gonioscopy, best-corrected visual acuity >20/30, no medical history of systemic disease, and no history of anti-glaucoma medication use before the initial clinic visit. At least eight IOP measurements using GAT at regular clinic visits were required for inclusion. Patients with glaucoma who underwent ophthalmic surgeries other than LRS (e.g., cataract surgery, vitrectomy, or glaucoma surgery) were excluded. Patients who discontinued using anti-glaucoma medication during the follow-up period(s) and those who used steroid or non-steroidal anti-inflammatory eye drops were also excluded. To minimise the floor effect of OCT measurements [14], we excluded eyes with a mean deviation (MD) of <−20 dB or peripapillary retinal nerve fibre layer (RNFL) thickness of <60 μm. If both eyes were eligible for inclusion, one eye was randomly selected.

All participants underwent complete ophthalmic examinations: measurements of best-corrected visual acuity, IOP using GAT, CCT measurements using an ultrasonic pachymeter (DGH-1000; DGH Technology, Inc., Frazer, PA, USA), slit-lamp biomicroscopy, dilated fundus examination, colour disc photography (Carl Zeiss Meditec, Jena, Germany), spectral-domain OCT (Cirrus HD-OCT, software v11.0; Carl Zeiss Meditec), axial length measurements (IOL Master; Carl Zeiss Meditec), and VF tests (Humphrey Field Analyzer II; Carl Zeiss Meditec).             

4. Provide on discussion some clinical practice advises to improve this technology in dairy ophthalmology routine.

-> We revised the conclusion section to emphasize the importance of IOP fluctuation and disc hemorrhage when managing glaucoma patients with a history of laser refractive surgery. We believe this would help ophthalmologists in their clinical environment.

-> Lines 315-317. These results suggest that when managing glaucoma patients with a history of LRS, IOP fluctuation should be minimized, and more aggressive treatment should be considered if DH is found.

5. Please clarify the conclusion to improve comprehension. Resume into 3 or 4 bullets sentence and provided shortened sentences.

-> We modified the conclusion paragraph to improve comprehension.
-> Lines 313-317. Risk factors for structural progression of open-angle glaucoma differed between the LRS and non-LRS groups. IOP fluctuation and DH were significant risk factors in glaucomatous eyes with a history of LRS. These results suggest that when managing glaucoma patients with a history of LRS, IOP fluctuation should be minimized, and more aggressive treatment should be considered if DH is found.

Reviewer 2 Report

I read this study with great interest.

I have few comments.

  1. It appeared that the average age of study population was much younger than what is expected in the glaucoma cases
  2. Were you adjusted for refractive error whilst acquiring the OCT images in non LRS group?
  3. It seems like all cases had mild to moderate glaucoma only. Can you elaborate on this?
  4. Can you explain why the posterior pole scan and GCL complex OCT were not studied?
  5. What was the definition of RNFL OCT progression? I appreciate that the progression analysis provided by the OCT software was utilised but what threshold was used?
  6. One of the limitations of the present study is the variability in IOP measurement by different operators

Author Response

I read this study with great interest. I have few comments.

1. It appeared that the average age of study population was much younger than what is expected in the glaucoma cases

-> We appreciate your positive feedback and hope that the revised manuscript is now suitable for publication.
-> The goal of our study is to compare risk factors of glaucoma progression between eyes with a history of laser refractive surgery and those without a history of laser refractive surgery. Since laser refractive surgery is more prevalent in the younger population, the group of glaucoma patients with  history of surgery  were younger than those investigated in other studies. To minimize the effect of age on glaucoma progression, we excluded individuals older than 60 years in both groups.

-> Lines 102-104. Because our preliminary data showed that the LRS group was younger than the non-LRS group, we only included individuals younger than 60 years to minimise the effect of age on structural progression.  

2. Were you adjusted for refractive error whilst acquiring the OCT images in non LRS group?

-> Good question. . Due to myopia, there could be an issue that the true diameter of the fundus increases with increasing axial length, known as the magnification effect. However, since the purpose of this study was to compare the risk factors between LRS group and non-LRS group with similar axial length, we didn’t think that adjustment for refractive is necessary. 

3. It seems like all cases had mild to moderate glaucoma only. Can you elaborate on this?

-> In advanced glaucoma eyes, structural change is already progressed so much that further change is not detected easily. Therefore, we excluded the advanced glaucoma eyes.

-> Lines 75-77. To minimise the floor effect of OCT measurements [14], we excluded eyes with a mean deviation (MD) of <−20 dB or peripapillary retinal nerve fibre layer (RNFL) thickness of <60 μm.

4. Can you explain why the posterior pole scan and GCL complex OCT were not studied?

-> Very great point. When we first designed this study, analysis for peripapillary retinal nerve fiber layer was only included, since this method has been used a lot to analyze the structural progression of glaucoma. However, as you mentioned, recent studies reported that the analysis for macular GCIPL could be useful in highly myopic eyes. So, we added sentences about this in the limitation section.

-> Lines 301-305. Additionally, only peripapillary RNFL was analysed to detect structural progression in this study. One recent study reported that monitoring of macular ganglion cell-inner plexiform layer measurements could be effective for predicting glaucoma progression in high myopia [35]. Therefore, we believe that further study including macular ganglion cell-inner plexiform layer analysis is needed.  

5. What was the definition of RNFL OCT progression? I appreciate that the progression analysis provided by the OCT software was utilised but what threshold was used?

-> We used guided progression algorithm provided in Cirrus OCT. Detailed method is described as follows.
-> Lines 116-127. We evaluated the structural progression of the peripapillary RNFL using an event-based algorithm provided by guided progression analysis. The guided progression analysis algorithm compared the changes in peripapillary RNFL thickness at individual superpixels (1 superpixel = 4 × 4 pixels) between two baseline thickness maps and the follow-up thickness map. For the change to have been considered significant, a change of at least 20 adjacent superpixels must have been detected in the RNFL thick-ness maps. If a follow-up OCT examination demonstrated a statistically significant difference in thickness that exceeded the baseline test–retest variability, the superpixel was labelled yellow to indicate possible loss. If confirmed on a second follow-up OCT examination, it was labelled red to indicate probable loss. Structural progression was defined as the detection of “likely loss” in the event analysis, and this change was observed at the most recent follow-up visit.

6. One of the limitations of the present study is the variability in IOP measurement by different operators

-> Agree. We modified the limitation section.

-> Lines 310-315. Forth, due to the retrospective nature of this study, IOP was measured by different examiners. The accuracy of IOP measurements may be affected by the inter-examiner differences. Despite this limitation, IOP of the participants in this study were measured using a consistent and standardized method, so inter-examiner differences for IOP measurements were considered controlled.
